# Stifle Joint Arthrodesis for Treating Chronic-Osteoarthritis-Affected Dogs

**DOI:** 10.3390/vetsci10070407

**Published:** 2023-06-21

**Authors:** Shin-Ho Lee, Yoon-Ho Roh, Dong-Bin Lee, Jae-Hyeon Cho, Chung-Hui Kim

**Affiliations:** 1Department of Companion Animal Care, Kyungnam College of Information & Technology, Busan 47011, Republic of Korea; hovet519@naver.com; 2Institute of Animal Medicine, College of Veterinary Medicine, Gyeongsang National University, Jinju 52828, Republic of Korea; yoonhoroh@gnu.ac.kr (Y.-H.R.); dlee@gnu.ac.kr (D.-B.L.); jaehcho@gnu.ac.kr (J.-H.C.)

**Keywords:** osteoarthritis, stifle joint arthrodesis, medial patellar luxation, lameness

## Abstract

**Simple Summary:**

Chronic joint pain is a common problem among dogs, for which owners often seek veterinary help. Affected dogs typically present painful lameness, stiffness, and severe atrophy of the muscles. The present study reports the case of a dog that was presented to a veterinary hospital due to lameness, osteoarthritis, and pain caused by patellar luxation, making it impossible for the dog to use its legs or walk. On the right hindlimb, we conducted stifle joint arthrodesis (SJA) and performed reconstructive surgery to correct medial patellar luxation (MPL) on the left hindlimb. For dogs unable to stand due to severe osteoarthritis caused by side effects after the dislocation of the medial patella, stifle joint arthrodesis results in the restoration of functional walking. This is surgical method can improve a dog’s quality of life by recovering its ability to perform movements required in many aspects of daily life.

**Abstract:**

A two-year-old male Pomeranian dog was presented to a veterinary hospital due to the side effects of a surgical correction for patellar luxation. Stifle joint arthrodesis (SJA) was performed on the patient’s right leg using autologous bone-grafting techniques. The right femur and tibial joint were angled 120–130°, and an SJA plate was fixed on the front of the two bones. After performing joint fusion of the right limb, medial-patellar-luxation-(MPL)-corrective surgery was performed to cut the tibial tuberosity on the left leg, and the fixing force was increased using the figure-of-eight-tension-band-wiring technique. Results were recorded regarding the dog’s ability to walk and trot in the right hind limb; these results were evaluated for 27 days after surgery. It was difficult for the patient to walk because weight-bearing had not been carried out for 3 days after the surgery; short strides and partial weight bearing were possible 5 to 7 days after surgery. After 10 days, the patient was able to move while bearing weight with a slight disruption. With regard to trotting, the patient showed intermittent normal steps 5 to 7 days after surgery, but the disruption continued. After 14 days, trotting was possible, and it was observed that movement could be maintained during everyday activities.

## 1. Introduction

Stifle joint arthrodesis (SJA) is one of the best treatment methods for highly virulent infections, loss of stifle flexion/extension, and weakened joints due to bone loss. SJA can be performed using various fixing methods, such as external fixation, plate internal fixation, and intramedullary bone fixation [1,2,3]. Recently, autologous cancellous bone grafts were used in cases of bone loss or where removal was required due to excessive inflammatory damage to cartilage or bone in joints [4]. Cancellous bones are also transplanted to supply various forms of proteins and cells for septic orthopedic injury treatment [5]. There are cases in which joint fusion was carried out due to end-stage elbow joint disease [6]. Three dogs are suffering from different conditions - trauma, osteomyelitis, and immune-mediated disease. Metatarsophalangeal arthrodesis was performed to treat end-stage degenerative joint disease [7].

The restriction of movement of the stifle joint and lameness can be induced by cartilage damage or inflammation caused by stifle joint luxation, the latter of which has a prevalence of 10–20% in dogs [8,9]. After surgery, it was reported that 55% of dogs suffered damage to the stifle cartilage, 26.3% of which were cruciate ligament ruptures, 43% were minor complications, and 18% were severe complications [10]. In the event of severe complications accompanied by cartilage damage after medial patellar luxation (MPL), SJA can be an effective surgical method because patients’ quality of life would otherwise deteriorate due to severe pain and sensory abnormalities.

This case concerns a 2-year-old neutered male Pomeranian dog that underwent two previous operations to correct MPL 5 months before the presentation but failed to recover and thus visited our animal hospital. This study attempted to evaluate the degree of recovery by measuring the preoperative and postoperative lameness scores after performing SJA on patients with severe chronic osteoarthritis and lameness due to complications related to repeated reoperation.

## 2. Materials and Methods

### 2.1. History and Presurgery

A 2-year-old, 3.9 kg, neutered male Pomeranian dog was brought to the animal hospital. The dog was diagnosed with congenital medial patella luxation at another local hospital, and after two failed corrective surgeries, he was brought to our hospital. Normal gait was impaired because the patella luxated continually and could not be manually replaced. The degree of luxation of the patella was grade Ⅳ [11].

It is not possible to bear continuous weight on the stifle joints in both hind limbs. The dog had difficulty using both hindlimbs due to inflammation and pain caused by the failure of bilateral MPL-corrective surgeries performed five months prior. Upon inspection and palpation, there was a dislocation in the left patella of the hind limb. The tibial tuberosity was internally rotated, and the right patella was not detected upon palpation. The patient had disuse muscular atrophy in both hind limbs due to lameness in the stifle joint, and a local heat sensation was reported around the stifle joint. Pain and discomfort were elicited during flexion and extension, and muscular atrophy was confirmed in both hind limbs. The patient preferred to have its knee flexed.

X-ray (E7239X, TOSHIBA, Tokyo, Japan) analysis was conducted. According to radiograph imaging (Weview pacs system, Seoul, Republic of Korea), the screw fixing the bone fragment to the tibial tubercle in the right hind limb had remained, and the patella was not recognized in the radiography view, which was consistent with the results following palpation (Figure 1A,B). Bone proliferation was noted, and edema was observed around the stifle joint (Figure 1A,B). Severe MPL, periostitis, and edema around the stifle joint in the left hind limb were observed (Figure 1A,C). Following a physical examination, complete blood count (Procyte Dx analyzer, Maine, USA) and biochemistry device (Catalyst One chemistry analyzer, Maine, USA) were measured. Neutrophil levels were observed to be slightly increased, namely, 11.71 μL (78.2%), suggesting that inflammatory responses were in progress (Table 1).

The MPL procedure was performed on the left stifle joint because the patella was present, the joint cartilage was not severely damaged, and the cranial cruciate ligament was not ruptured. In contrast, it was decided to perform the SJA procedure on the right leg because there was no patella, the cruciate ligament was ruptured, and the level of arthritis was severe.

Before surgery, antibiotic and anti-inflammatory drugs were applied for about a month to relieve pain, edema, and arthritis; then, the surgery was performed. For treatment, amoxicillin/clavulanic acid (12.5 mg/kg, Amocla, KUHNIL corp., Seoul, Republic of Korea), carprofen (2.2 mg/kg, RIMADYL, Zoetis Inc., Parsippany-Troy Hills, NJ, USA), tramadol (2 mg/kg, Tridol cap. Yuhan Corp, Seoul, Republic of Korea), and famotidine (0.5 mg/kg, Famotidine, Hanmi Pharm, Seoul, Republic of Korea) were prescribed BID for 4 weeks. After one month, radiographic findings confirmed that inflammation and edema in both stifle joints were reduced (Figure 2). The owner was informed that stifle joint arthrodesis was to be performed on the right hind limb due to severe osteoarthritis, loss of the patella, and rupturing of the cranial cruciate ligament and that MPL correction was to be performed on the left leg. Accordingly, the consent of the guardian was obtained.

### 2.2. Preoperative Preparation

Before surgery, the surgical site was shaved to create a sterile area, and Betadine (Povidone iodine, Sungkwang Pharm., Bucheon, Republic of Korea) was applied for 20 min to sterilize the site. Then, Betadine was removed with 70% alcohol, and the surgical area to be operated on was thoroughly disinfected. Next, a sterile surgical cloth was placed on the affected area to prevent contamination. Cefazolin (25 mg/kg, Cefazolin Inj. Chong Kun Dang Pharm, Seoul, Republic of Korea) was injected intravenously. Butorphanol (0.1 mg/kg, Myungmoon pharm co., Seoul, Republic of Korea) was administered as premedication for anesthesia. Propofol (10 mg/kg, Provive, Myungmoon Pharm, Seoul, Republic of Korea) was injected intravenously as an induction agent for tracheal intubation, and the patient was positioned in dorsal recumbency for the surgery. After intubation using an endotracheal tube (Rushelit, size ID 3.5 mm, OD 5.3 mm, Teleflex, Kamunting, Malaysia), general anesthesia, using isoflurane (Ifran, Hana Pharm, Republic of Korea), was induced by forced breathing using a respiratory anesthetic machine (Drager primus, Dragerwerk AG & Co. KGaA, Lübeck, Germany) with a volume of 50 to 60 cc. In addition, normal saline (Saline 100 mL Bag, Sejung, Seoul, Republic of Korea) was administered at an infusion rate of 5 mL/kg/h intravenously to maintain adequate perfusion and circulation during surgery.

### 2.3. Stifle Arthrodesis

A skin incision in the right stifle joint for SJA was made anteriorly between the right femur and tibia. The skin and fascia were cut into 15 cm longitudinal sections, and the fascia was isolated by approaching the medial side of the quadriceps femoris muscle (Figure 3A).

After incising the articular capsule, the femur and tibia were exposed by rotating the joint capsule 180° laterally using Allis tissue forceps (23 cm, KASCO, Islamabad, Pakistan) (Figure 3B). A rupture of the cranial cruciate ligament was confirmed in the process of exposing the stifle joint. The tissue attached around the stifle joint was partially removed, and fibrous adhesive tissue that interfered with flexural movement was separated and removed (Figure 3C). The distal end of the femur and the proximal end of the tibia were flattened with a bone file and rasp (Bone rasp and File, Professional, Sialkot, Pakistan) and a rongeur (Single bone rongeur, 12 cm, Mabson industry, Sialkot, Pakistan), allowing the plate and the joint surface to be attached. Cancellous bone was extracted during the process of flattening the joint surface. It was stored in an empty syringe to avoid contamination and served as an autologous graft for improving bone union. In order to fix the stifle joint at an angle of 120–130°, using a K-wire (1.4 mm × 229 mm, General Vet Products, Fairy Meadow, Australia), a guide pin was inserted through the medial surface of the tibia in an upward direction of 45° with respect to the medial condyle of the femur. In the same way, the stifle joint was fixed using two guide pins positioned in an ‘x’ shape (Figure 4). Cancellous bone tissue was transplanted into the stifle joint cavity. The angle and location of the plate (ID = 2.0 mm and length = 8.5 cm, Orthotech, Daegu, Republic of Korea) for SJA were confirmed using a C-arm machine (7700 Compact C-Arm, Hi Tech International Group Inc., Deerfield Beach, FL, USA) and fixed on the front of the femur and tibia. The screws used for fixing the plate were 1.5 mm and 2.0 mm in size (Veterinary Instrumentation, Doiff, Suncheon, Republic of Korea) and inserted from the proximal part of the femur to the distal part and, alternatively, from the distal part of the tibia to the proximal part (Figure 4A,B).

### 2.4. Reconstruction of MPL

After performing SJA of the right hind limb, MPL correction of the left hind limb was performed. The approach was carried out by making a vertical incision of about 10 cm in the craniolateral parts around the stifle joint. After the fascia and articular capsule were incised to expose the trochlea of the femur, tracheoplasty was performed in a wedge shape to secure the depth of the trochlea. Then, the wedge-shaped articular cartilage was inserted into the femoral trochlea. Subsequently, a tibial tuberosity that had moved toward the inside was cut using a bone saw and moved laterally to fit the patella into the trochlear groove using a K-wire (Figure 4A,C). Stability was further enhanced using the figure-of-eight-tension-band-wiring technique. Before suturing, the distal tibia was rotated internally and externally 2–3 times to confirm that the patella (Figure 4A,C) remained stable in the newly formed trochlear groove. Subsequently, routine closure was performed in the following order: joint capsule, muscle, subcutaneous tissue, and skin.

### 2.5. Postoperative Management

After SJA and MPL surgery, cephalexin capsules (25 mg/kg, Donghwa Pharm, Seoul, Republic of Korea), tramadol (2 mg/kg, Tridol cap. Yuhan Corp, Seoul, Republic of Korea), and carprofen tablets (2.2 mg/kg, Zoetis Inc., Parsippany-Troy Hills, NJ, USA) were prescribed BID for 10 days to prevent inflammation and control pain. During the hospitalization period after the surgery, a cold compress was applied to the stifle joint twice a day for pain management, and proprioceptive exercises were performed on a balance ball for 5–10 min twice a day to prevent muscle atrophy and balance loss. For the rest of the period, the patient was allowed to rest in a cage to promote adequate recovery. For 3 days following surgery, the dog’s lameness score was checked, and light walking was induced twice a day for 10–20 min (without a leash) to prevent muscular atrophy. Ten days after surgery, the patient was discharged and prescribed carprofen 4.4 mg/kg SID for 20 days for pain relief because there were no evident problems.

### 2.6. Lameness Score

The lameness score was evaluated on postoperative days 3, 5, 7, 10, 14, 17, 22, and 27 while the dog walked and trotted on its right hind limb to assess the degree of recovery. The evaluation was performed using the method proposed by Millis and Levine [11] (Table 2). Walking and trotting were evaluated using video recordings. Walking is characterized by a four-beat gait, while a diagonal, fast, two-beat gait is classified as trotting.

### 2.7. Results

After performing SJA and MPL surgery on the patient, the patient was observed while walking using its right hind limb for 27 days, and the results are shown in Figure 5. For 3 days after surgery, the patient’s walking ability corresponded to grade 5, and a continuous weight load could not be carried. From days 5 to 7, the dog’s walking ability improved to grade 3, it was able to take short steps, and partial weight bearing was possible (Figure 5A). From days 14 to 27, the patient’s walking ability improved to grade 1 with slight lameness, but the patient was able to bear weight and move around (Figure 5A). In the trotting analysis, the dog could not bear its weight, and its trotting ability corresponded to grade 5 for the first 3 days after surgery and grade 2 from days 5 to 7, presenting occasional normal trotting but still showing lameness (Figure 5B). From days 10 to 27, trotting improved to grade 1, with slight lameness, but the dog was able to trot normally (Figure 5B).

## 3. Discussion

Among dogs and cats, stifle osteoarthritis causes pain, reduces activity levels, and causes lameness orthopedically [12]. The most common cause of stifle osteoarthritis is a luxating patella (LP), but cranial cruciate ligament rupture (CCLR) is one of the main causes of severe stifle osteoarthritis [13,14]. The cranial cruciate ligament is the most important element in maintaining and protecting the stifle joint function among dogs and cats. CCLR is more likely to occur in large dogs with an active lifestyle than in small dogs, with a 5.5-fold higher probability, and for every 5 kg increase in body weight, the complication rate increases by 1.3 times. Among small dogs, it can also occur due to inflammation after LP surgery [10]. In particular, Pomeranians are the breed most affected by LP, accounting for 41.2% of cases evaluated from 1974 to 2012 [15]. SJA is applied when pain and dysfunction are severe or when it is impossible to employ other surgical options. Thus, SJA is performed as a surgical treatment to reduce pain and improve quality of life [16]. The lameness score is an essential clinical outcome assessment tool for gait analysis. Walking and trotting are the most commonly evaluated gaits because they are symmetrical and can easily indicate lame limbs, making it easy to assess the abnormality of moving limbs without specialized equipment [17].

This is a case report on the use of SJA on a 2-year-old Pomeranian who underwent two surgeries for MPL treatment but did not fully recover and developed CCLR, joint inflammation, hind limb lameness, and persistent pain. SJA is a surgical method for alleviating pain and maximizing joint mobility by connecting joint surfaces together and then fixing them with plates and screws, allowing for normal daily life activities when complete recovery is achieved. This method is used in cases where the joint is damaged due to congenital abnormalities or inflammation caused by trauma, causing difficulty in maintaining normal joint function. The use of autologous cancellous grafts is recommended if there is a great deal of damage to the joint area or if recovery from septic damage is required [6]. In this study, the patient presented CCLR, severe pain, inflammation in the stifle joint, and loss of joint function. Therefore, we performed SJA with autogenous cancellous grafting on the right limb and MPL-corrective surgery on the left limb. Gait analysis was conducted to observe the results regarding walking and trotting through the lameness score after surgery. Before the surgery, the patient was unable to walk on its right hind limb and relied on its left hind limb for daily activities. During palpation of the hind limb area, the patient screamed due to pain and could not be treated. However, three days after surgery, the pain had almost disappeared, and the patient tried to bear weight on the right hind limb with some intermittent stepping. By the fifth day, the patient was observed walking with obvious lameness. After 10 days, walking was possible at grade 1, and the patient showed only a slight degree of lameness that did not interfere with daily activities. However, there was compensatory abnormal walking, and trotting reached grade 1 after 14 days, which was 4 days later than walking.

Since the stifle joint was fixed using SJA, a compensatory gait was observed during walking and trotting, for which the hip and hock joints were used. This was not a normal form of movement, but it showed a circumduction gait in the form of a line motion. We concluded that there were no further after-effects because the stifle joint did not show any signs of instability, and there was no swelling or inflammation a month after surgery. When autogenous cancellous grafting using the metaphyses of the humerus was performed with elbow joint arthrodesis, successful fusion was observed 6 weeks after surgery, with some lameness remaining, but stable fixation of the joint and improvement in walking were observed, which constitute results similar to those of reported in [18].

Since dogs have four limbs on the ground and have different weight distributions, there is a difference in the proportion of weight loads on the front and rear limbs [11]. In general, dogs support about 30% of their body weight on each of the cranial limbs and 20% on each of the caudal legs in a standing position. Regarding the difference between walking and trotting, walking speeds in medium-sized and large dogs range from 0.7 to 1.0 m/sec, during which about 60% of the weight is applied to the forelimbs and 40% to the hind limbs. In the case of trotting, the speed ranges from 1.7 to 2.0 m/sec, and the weight bearing is 100 to 120% on each front limb and 65 to 70% on each hind limb [11]. If a dog moves to reduce weight bearing on a limb, the weight bearing on the other limbs increases. If this condition persists for a long time, a limb that is under constant load can cause joint problems, so it is necessary to treat the painful limb as soon as possible.

In this case, the patient was unable to use its right hind limb due to the side effects of two stifle joint surgeries; however, walking and trotting were possible about 14 days after the SJA and MPL operations, thereby reducing the weight bearing on the left hind limb. The lameness score shown in Figure 5 is a widely used tool in dog lameness evaluation because walking and trotting can be easily evaluated without specialized equipment. Since dogs walk symmetrically, their unique walking can be easily distinguished visually. Regarding the measurement of lameness, the degree of lameness is insufficiently confirmed while walking due to the low amount of force applied, but the degree of lameness increases when trotting, which induces a higher amount of force when compared to walking. Therefore, to evaluate walking in detail, both walking and trotting gaits should be examined.

This study confirmed improvements using only the owner’s questionnaire and the lameness score. There is a limitation in that the improvement effect of the rehabilitative intervention was not evaluated using kinetics and kinematics. It is thought that the effect of SJA on chronic osteoarthritis can be clarified through various evaluation methods.

## 4. Conclusions

In the present study, we evaluated the degree of recovery through walking and trot-ting lameness scores after SJA in a patient that could not use hind limbs due to the side effects of MPL-corrective surgery. Through future technological advances, it is hoped that SJA will be widely used as a surgical method for recovery among animals suffering from joint pain.

## Figures and Tables

**Figure 1 vetsci-10-00407-f001:**
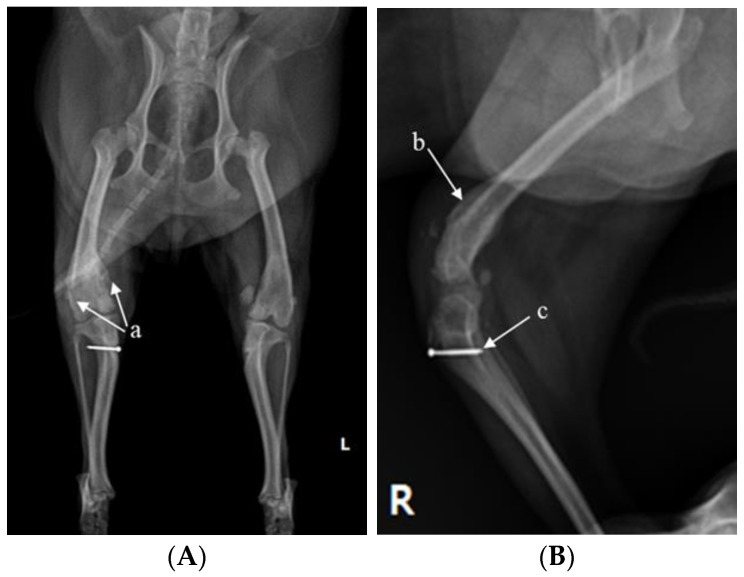
A 2-year-old Pomeranian dog with chronic weight-bearing lameness of the stifle joints in both hind limbs. (**A**) Craniocaudal radiographic views of the stifle joints of the dog. (**B**) Lateral radiographic views of the right stifle joints. (**C**) Medial radiographic views of the left stifle joints. a: osteophyte, b: periostitis, c: residual screw for tibial tuberosity transposition, d: periostitis, R: right, and L: left.

**Figure 2 vetsci-10-00407-f002:**
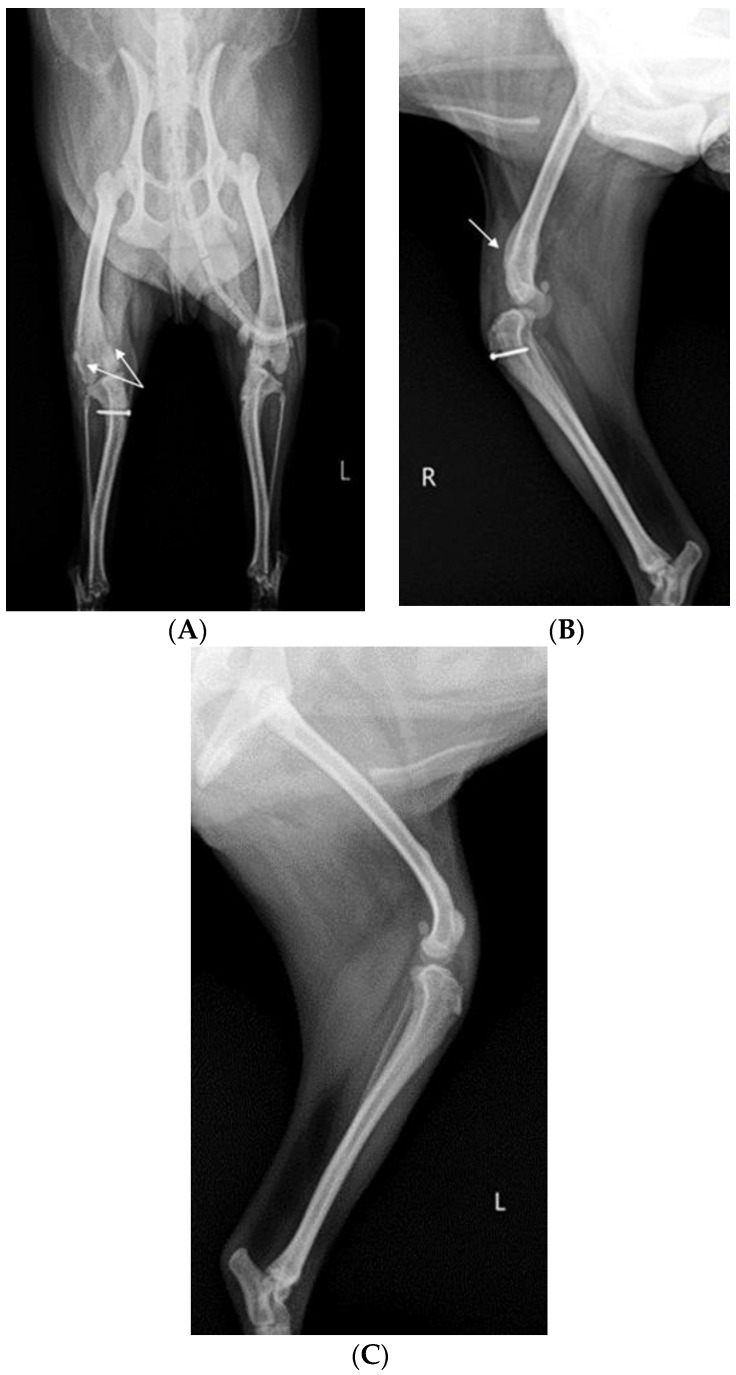
Preoperative craniocaudal (**A**) and mediolateral (**B**,**C**) radiographs of the right and left stifle joints. The prescribed anti-inflammatory drugs reduced pain, lowered fever, and decreased inflammation (swelling and damage) after about a month. Arrows in images show areas with significantly reduced inflammation and edema compared to a month prior.

**Figure 3 vetsci-10-00407-f003:**
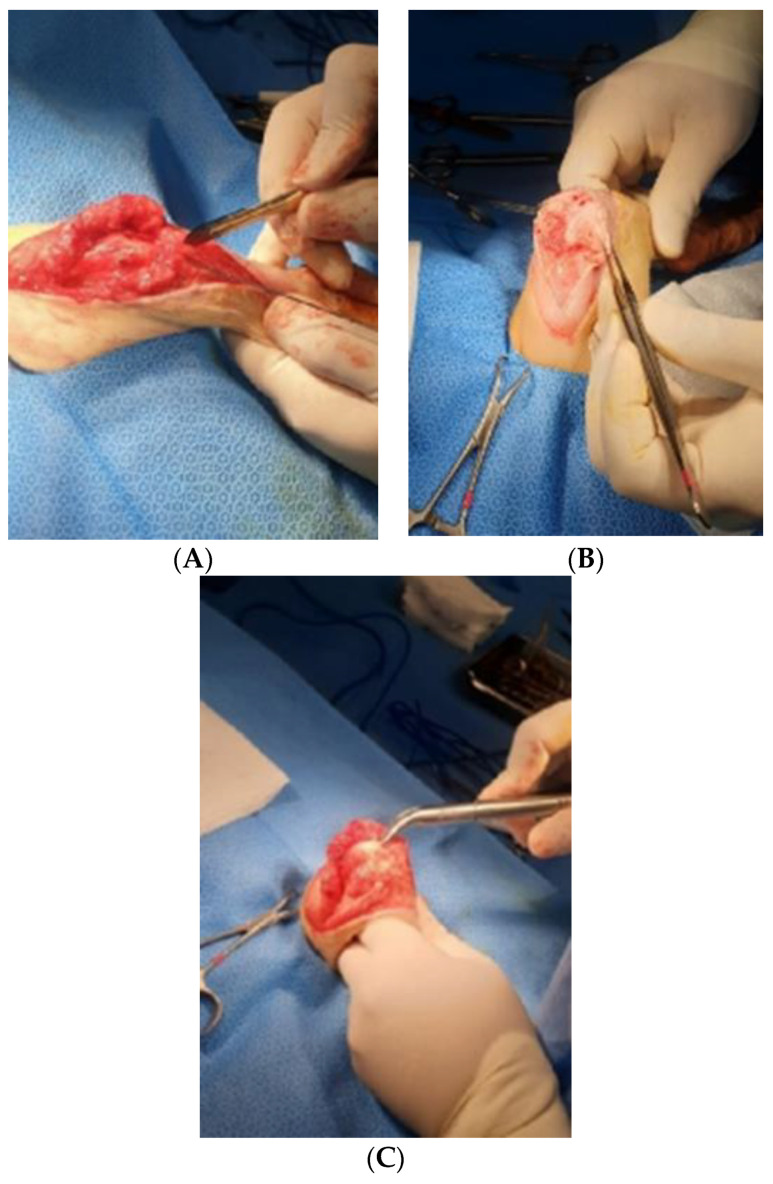
Intraoperative images of the stifle joint. (**A**) The process of removing the attached tissue around the joint to expose the stifle joint. (**B**) The process of exposing the stifle joint and identifying the cranial cruciate ligament injury and degenerative arthritis. (**C**) The process of removing adhesive tissue for stifle joint arthrodesis.

**Figure 4 vetsci-10-00407-f004:**
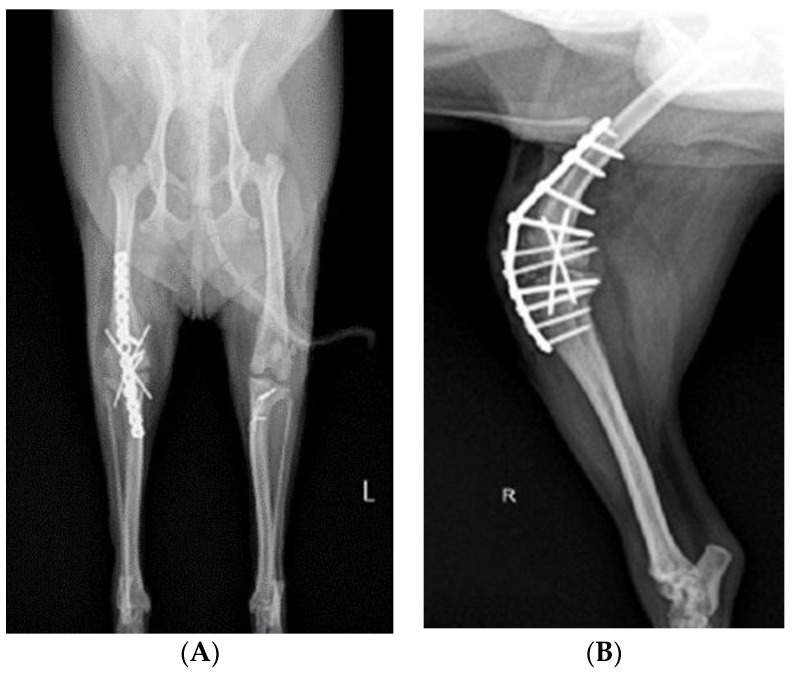
Radiograph images depicting the stifle joint arthrodesis performed on the right hind limbs and the medial patellar luxation correction of the left hind leg. (**A**) Craniocaudal radiographic view of both stifle joints. (**B**) Mediolateral radiographic view of stifle joints following SJA. (**C**) Postoperative mediolateral view of MPL in the left hindlimb.

**Figure 5 vetsci-10-00407-f005:**
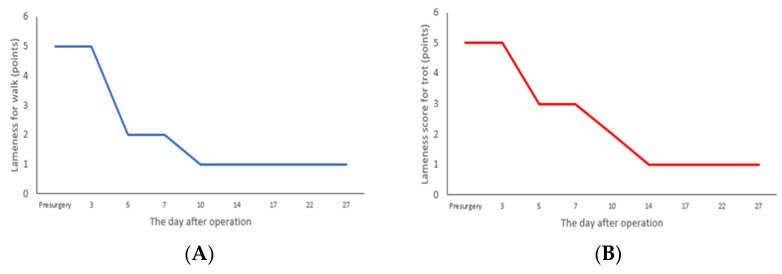
Variation in values of lameness score after SJA and MPL operations. (**A**) Walking assessment. (**B**) Trotting assessment.

**Table 1 vetsci-10-00407-t001:** Complete blood cell count and blood biochemistry.

Description	Result	Unit	Reference Range
HCT	43.4	%	37.3~61.7
HGB	14.6	g/dL	13.1~20.5
RBC	7.57	10^12^/L	5.65~8.87
MCH	19.3	pg	21.2~25.9
MCHC	33.6	g/dL	32~37.9
MCV	57.3	fL	61.6~73.5
RDW-CV	23.2	%	13.6~21.7
RETIC	57.5	10^3^/μL	10~110
PLT	494	10^9^/L	148~484
MPV	14.2	fL	8.7~13.2
PCT	0.7	%	0.14~0.46
WBC	14.99	10^9^/L	5.05~16.76
WBC-BASO	0.02	10^9^/L	0~0.1
WBC-EOS	0.66	10^9^/L	0.06~1.23
WBC-LYM	2.1	10^9^/L	1.05~5.1
WBC-MONO	0.5	10^9^/L	0.16~1.12
WBC-NEU	11.71	10^9^/L	2.95~11.64
RETHGB	22.3	K/uL	22.3~29.6
ALT	85	U/L	10~125
ALKP	82	U/L	23~212
BUN	15	mg/dl	7~27
CREA	0.6	mg/dl	0.5~1.8
BUN/CR	27	mg/dl	6~20
TP	7.4	g/L	5.2~8.2
ALB	3.4	g/L	2.3~4
GLOB	4	g/L	2.5~4.5
GLU	82	mg/dl	74~143

**Table 2 vetsci-10-00407-t002:** Lameness evaluation while walking and trotting.

Score	Assessment
**Lameness Evaluation at a Walking**	
0	Walks normally
1	Slight lameness
2	Obvious weight-bearing lameness
3	Severe weight-bearing lameness
4	Intermittent non-weight-bearing lameness
5	Continuous non-weight-bearing lameness
**Lameness Evaluation at a Trot**	
0	Trots normally
1	Slight lameness
2	Obvious weight-bearing lameness
3	Severe weight-bearing lameness
4	Intermittent non-weight-bearing lameness
5	Continuous non-weight-bearing lameness

## Data Availability

Not applicable.

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
