# Peer review of "Stifle Joint Arthrodesis for Treating Chronic-Osteoarthritis-Affected Dogs"

_vetsci, 2023, doi:10.3390/vetsci10070407_

Round 1

Reviewer 1 Report

This paper concerns an arthrodesis procedure following a failed surgical attempt to correct a patellar dislocation. This is valuable because papers describing the management of complications are not common. I am puzzled by the anasthetic proto-collection. Can the authors explain why they did not use preoperatively and/or intraoperatively some opioid analgesic such as fentanyl. The procedure carried out was invasive and painful.
Why was carprofen used and not, for example, robenacoxib or cimicoxib?

[60] it can be assumed that the dislocation of the patella was congenital and not post-traumatic or please write this for clarity of the case description.

{60} please specify the degree of luxation of the patella which was the cause of surgery (eg. https://www.ncbi.nlm.nih.gov/pmc/articles/PMC6055913/  )

[71] please specify model of x-ray machine and software

[77] please indicate which equipment was used for the blood tests

[80] Is it the cranial cruciate ligament? Please specify.

[270] please replace legs with limbs

Author Response

Prof. Dr. Patrick Butaye

Editor-in-Chief

Department of Pathobiology, Pharmacology and Zoological Medicine, Faculty of Veterinary Medicine, Ghent University, Salisburylaan 133, 9820 Merelbeke, Belgium

Manuscript ID: vetsci-2395224

Type of manuscript: Case Report

Title: Stifle joint Arthrodesis for Treating Chronic Osteoarthritis in a Dog.

Dear Dr. Patrick Butaye

Thank you for your comments on our manuscripts entitled, “Stifle joint Arthrodesis for Treating Chronic Osteoarthritis in a Dog”. We have carefully revised the manuscript taking into account all suggestions by the reviewers. Please find below a detailed point-by-point response to all comments (reviewers’ comments in Q, our replies in A). We would like to thank the reviewers for their efforts and time, particularly, for useful comments to make our paper better. We believe that our manuscript has been substantially improved by this revision.  

Thank you very much for your time and consideration.

Your Sincerely,

Chung Hui Kim, D.V.M. Ph.D

Jiju Daero 501, Jinju-city,

Gyeongsangnamdo, 52828, Korea

College of Veterinary Medicine

Gyeongsang National University

Phone: +82-55-772-2375

Fax: +82-55-772-2349

Email: kimchi3237@gnu.ac.kr

Comments from Reviewer 1

(Q) This paper concerns an arthrodesis procedure following a failed surgical attempt to correct a patellar dislocation. This is valuable because papers describing the management of complications are not common. I am puzzled by the anasthetic proto-collection. Can the authors explain why they did not use preoperatively and/or intraoperatively some opioid analgesic such as fentanyl. The procedure carried out was invasive and painful. Why was carprofen used and not, for example, robenacoxib or cimicoxib?

(A) We highly appreciate the reviewers' insightful and helpful comments on our manuscript. The surgery is long and it's a very respectable part of pain management. We also believe that surgery is rarely the first choice when it comes to treating painful. Intraoperative and postoperative pain on the patients is a significant problem but there is no need for the patient to be in a lot of pain. Thus, pre or/and postoperative pain management is need to eliminate pain and discomfort with minimal side effects. We knew robenacoxib is a veterinary approved non‐steroidal anti‐inflammatory drug to treat dogs. However, robenacoxib and cimicoxib are not officially used in Korean hospital, and we couldn't apply them. In this study, tramadol (2mg/kg, q12hr, Tridol inj/cap. Yuhan Corp, Korea) was prescribed for each pre-and postoperative pain management. The patient's condition is very good after operation.

(Q) [60] it can be assumed that the dislocation of the patella was congenital and not post-traumatic or please write this for clarity of the case description.

(A) We appreciate reviewer’s comment. In addition, the dog was diagnosed with congenital medial patella luxation at another local hospital, and after two failed corrective surgeries, he was presented. Text is revised.

(Q) [60] please specify the degree of luxation of the patella which was the cause of surgery (eg. https://www.ncbi.nlm.nih.gov/pmc/articles/PMC6055913/  )

(A) We appreciate reviewer’s comment. The text is revised and add the reference as A 2-year-old, 3.9 kg neutered male Pomeranian dog visited the animal hospital with congenital MPL. Normal gait was impaired because the patella luxated continually and cannot be manually replaced. The degree of luxation of the patella was grade Ⅳ [18].

(Q) [71] please specify model of x-ray machine and software.

(A) We appreciate reviewer’s comment. Text is revised in Materials and Methods section.

(Q) [77] please indicate which equipment was used for the blood tests.

(A) We appreciate reviewer’s comment. Text is revised in Materials and Methods section.

(Q) [80] Is it the cranial cruciate ligament? Please specify.

(A) Thank you for pointing this out. Text is corrected as directed.

(Q) [270] please replace legs with limbs

(A) Thank you for this suggestion. Text is revised.

Reviewer 2 Report

General comments: 

This is a very interesting article about SJA for treating chronic osteoarthritis in a dog. However it is only one case, so I would suggest a more detailed clinical case presentation throughout the manuscript, namely with a detailed clinical examination, specific measurements and precise evaluation outcomes. It would be interesting to compare this results with other treatments and other possible surgeries. Also, there is a need to explore some matters in the discussion section. 

Specific comments: 

Line 44: The authors refer to arthrodesis being performed to treat osteoarthritis in horses. What about dogs? Is there any research? 

Line 49: The authors refer to severe complications? What are these severe complications? Please specify. 

Line 55: The authors talk about evaluating the degree of recovery? But how? I suggest to add some information about possible outcomes to evaluate this recovery.

Line 60: What is the score of lameness of this dog? 

Line 68: In regard to the joint range of motion, what was the specific range of motion in flexion and extension (goniometry measures were taken?). And muscle mass measurements? 

Line 86: There were 4 weeks of carprofen right? Was there any gastric protection made? This would be important to refer. Also to discuss in the discussion section, other type of medications that could be an option, such as cbd, gabapentin, tramadol and other examples? 

Line 121-130: What about pre-anesthesia? There was not any? Only propofol? What about Isoflurane CAM and fluid therapy infusion rate? Please specify.

Line 194: I suggest to the authors the use of photos in this paragraph, for example in regard to cold compression, proprioceptive exercises, ...

Results section: There was no long-term follow-up? This could be important for the study. Also, the kinematic and kinetic study of this case could be interesting, maybe to state this as a limitation. 

Discussion section: I suggest introducing more bibliographic references, mainly in the first paragraph of the discussion. Also to talk about the possibility of rehabilitation? This patient did physical rehabilitation, such as for muscle strengthening, laser class IV to reduce edema and inflammation? Why not? Please discuss this subject. 

Line 270: How was body weight distribution measured?

Author Response

Prof. Dr. Patrick Butaye

Editor-in-Chief

Department of Pathobiology, Pharmacology and Zoological Medicine, Faculty of Veterinary Medicine, Ghent University, Salisburylaan 133, 9820 Merelbeke, Belgium

Manuscript ID: vetsci-2395224

Type of manuscript: Case Report

Title: Stifle joint Arthrodesis for Treating Chronic Osteoarthritis in a Dog.

Dear Dr. Patrick Butaye

Thank you for your comments on our manuscripts entitled, “Stifle joint Arthrodesis for Treating Chronic Osteoarthritis in a Dog”. We have carefully revised the manuscript taking into account all suggestions by the reviewers. Please find below a detailed point-by-point response to all comments (reviewers’ comments in Q, our replies in A). We would like to thank the reviewers for their efforts and time, particularly, for useful comments to make our paper better. We believe that our manuscript has been substantially improved by this revision.  

Thank you very much for your time and consideration.

Your Sincerely,

Chung Hui Kim, D.V.M. Ph.D

Jiju Daero 501, Jinju-city,

Gyeongsangnamdo, 52828, Korea

College of Veterinary Medicine

Gyeongsang National University

Phone: +82-55-772-2375

Fax: +82-55-772-2349

Email: kimchi3237@gnu.ac.kr

Comments from Reviewer 2

(Q) Line 44: The authors refer to arthrodesis being performed to treat osteoarthritis in horses. What about dogs? Is there any research?

(A) We would like to thank the reviewer's comments. Text is revised as  in each of three dogs with trauma and osteomyelitis and immune-mediated disease, metatarsophalangeal arthrodesis was performed to treat end-stage degenerative joint disease [7].

(Q) Line 49: The authors refer to severe complications? What are these severe complications? Please specify.

(A) We appreciate reviewer’s comment. After surgery for complication following tibial tuberosity transposition in 137 canine stifles with medial patellar luxation, 55% of dogs suffered damage to the stifle cartilage, and 26.3% of these were cruciate ligament ruptures, 43% were of minor complications and 18% were of severe complications [10].

(Q) Line 55: The authors talk about evaluating the degree of recovery? But how? I suggest to add some information about possible outcomes to evaluate this recovery.

(A) We appreciate reviewer’s comment. Text is revised following reviewer ‘s suggestion as This study attempted to evaluate the degree of recovery by evaluating the preoperative and postoperative lameness score after applying SJA in patients with severe chronic osteoarthritis and lameness due to complications related to repeated reoperation.

(Q) Line 60: What is the score of lameness of this dog?

(A) We appreciate reviewer’s comment The dog visited the hospital due to lameness, osteoarthritis and pain caused patellar luxation, making it impossible for the dog to use both legs or walk. When the dog came to the hospital, the Lameness score was 5 in Trot and Walk.

(Q) Line 68: In regard to the joint range of motion, what was the specific range of motion in flexion and extension (goniometry measures were taken?). And muscle mass measurements?

(A) We understand the reviewer's viewpoint here, and we agree with the reviewer's advice that the measurement for the specific range of motion in flexion and extension could be logic and advantageous in this study. But we couldn’t evaluate ROM test with a goniometer due to Pain and vocalization. Also, since the dog was very anxious and disliked to be touched, we didn’t measure muscle circumference with Gulik.

(Q) Line 86: There were 4 weeks of carprofen right? Was there any gastric protection made? This would be important to refer. Also to discuss in the discussion section, other type of medications that could be an option, such as cbd, gabapentin, tramadol and other examples?

(A) We appreciate reviewer’s comment. The text is revised and added. famotidine (0.5 mg/kg, Famotidine, Hanmi Pharm, Korea) were prescribed bid for 4 weeks.

(Q) Line 121-130: What about pre-anesthesia? There was not any? Only propofol? What about Isoflurane CAM and fluid therapy infusion rate? Please specify.

(A) Thank you for your comments. Butorphanol was administered as premedication for anesthesia, and test is revised. In addition, normal saline was administrated at an infusion rate of 10ml/kg/h intravenously to maintain adequate perfusion and circulation during surgery.

(Q) Line 194: I suggest to the authors the use of photos in this paragraph, for example in regard to cold compression, proprioceptive exercises, ...

(A) We greatly appreciate the reviewer's valuable comment. Although we carried out the patient to do a proprioceptive exercises along with conventional physical therapy and the proprioceptive exercises were important in the treatment of musculoskeletal disorders, we couldn't take pictures because we was focused on surgery and recovery.

(Q) Results section: There was no long-term follow-up? This could be important for the study. Also, the kinematic and kinetic study of this case could be interesting, maybe to state this as a limitation.

(A) We appreciate reviewer’s comment. We recently talked to dog’s owner and confirmed that he is still doing well. It would have been better to use goniometer for the kinematics. The discussion section is revised as this study confirmed improvement only using the owner's questionnaire and lame-ness score. There is a limitation in that the improvement effect of rehabilitative interven-tion using kinematics and kinematics was not evaluated. It is thought that the effect of SJA in chronic osteoarthritis can be clarified through various evaluation methods.

(Q)  Discussion section: I suggest introducing more bibliographic references, mainly in the first paragraph of the discussion. Also to talk about the possibility of rehabilitation? This patient did physical rehabilitation, such as for muscle strengthening, laser class IV to reduce edema and inflammation? Why not? Please discuss this subject.

(A) We appreciate the reviewer's suggestion. The text is revised in discussion section as The lameness score is an essential clinical outcome assessment tool for gait analysis. The walk and trot are the most commonly evaluated gait because they are symmetrical and can easily identify lame limbs, making it easy to assess the abnormality of moving limbs without specialized equipment [11].

(Q)  Line 270: How was body weight distribution measured?

(A) We thank the reviewer for this helpful comment. We explained the weight distribution on the limbs in the text and added reference. To properly support the head, the front limbs bear a greater weight load compared to the rear limbs. Since dogs have four limbs on the ground and have different weight distributions, there is a difference in the proportion of weight loads on the front and rear limbs [11]. In general, dogs support about 30% of their body weight on each of the cranial limbs and 20% on each of the caudal limbs in a standing position.

Round 2

Reviewer 1 Report

The work can be published after revisions. I hope the new NSAIDs will be approved for animal use in Korea soon.

Reviewer 2 Report

This manuscript was improved after the revision and is ok for publication.